# New Insights into the Gut Microbiota in Neurodegenerative Diseases from the Perspective of Redox Homeostasis

**DOI:** 10.3390/antiox11112287

**Published:** 2022-11-18

**Authors:** Yu Wang, Zhe Zhang, Bowen Li, Bo He, Lei Li, Edouard C. Nice, Wei Zhang, Jia Xu

**Affiliations:** 1West China School of Basic Medical Sciences & Forensic Medicine, and State Key Laboratory of Biotherapy and Cancer Center, West China Hospital, Sichuan University, and Collaborative Innovation Center for Biotherapy, Chengdu 610041, China; 2School of Basic Medical Sciences, Chengdu University of Traditional Chinese Medicine, Chengdu 611137, China; 3Department of Biochemistry and Molecular Biology, Monash University, Clayton, VIC 3800, Australia; 4West China Biomedical Big Data Center, West China Hospital, Sichuan University, Chengdu 610041, China; 5Mental Health Center and Psychiatric Laboratory, the State Key Laboratory of Biotherapy, West China Hospital of Sichuan University, Chengdu 610000, China; 6School of Medicine, Ningbo University, Ningbo 315211, China

**Keywords:** neurodegenerative diseases, oxidative stress, redox homeostasis, gut microbiota, microbiota–gut–brain axis

## Abstract

An imbalance between oxidants and antioxidants in the body can lead to oxidative stress, which is one of the major causes of neurodegenerative diseases. The gut microbiota contains trillions of beneficial bacteria that play an important role in maintaining redox homeostasis. In the last decade, the microbiota–gut–brain axis has emerged as a new field that has revolutionized the study of the pathology, diagnosis, and treatment of neurodegenerative diseases. Indeed, a growing number of studies have found that communication between the brain and the gut microbiota can be accomplished through the endocrine, immune, and nervous systems. Importantly, dysregulation of the gut microbiota has been strongly associated with the development of oxidative stress-mediated neurodegenerative diseases. Therefore, a deeper understanding of the relationship between the gut microbiota and redox homeostasis will help explain the pathogenesis of neurodegenerative diseases from a new perspective and provide a theoretical basis for proposing new therapeutic strategies for neurodegenerative diseases. In this review, we will describe the role of oxidative stress and the gut microbiota in neurodegenerative diseases and the underlying mechanisms by which the gut microbiota affects redox homeostasis in the brain, leading to neurodegenerative diseases. In addition, we will discuss the potential applications of maintaining redox homeostasis by modulating the gut microbiota to treat neurodegenerative diseases, which could open the door for new therapeutic approaches to combat neurodegenerative diseases.

## 1. Introduction

Progressive neuronal necrosis and degeneration are hallmarks of neurodegenerative diseases (NDs), including Alzheimer’s disease (AD), Parkinson’s disease (PD), Huntington’s disease (HD), and multiple sclerosis (MS), which are caused by neurotoxic etiological agents in the brain and surrounding organs. Because of the diverse and complex pathological symptoms, uncertain pathogenesis, restricted clinical examination, difficulty in making an early diagnosis, and lack of treatment options, NDs have imposed significant societal and economic burdens [1,2,3,4]. The current treatment of NDs mainly focuses on relieving symptoms, and no effective cure is currently available. Therefore, the search for novel and effective strategies to treat NDs remains a priority.

As a result of normal metabolic processes, reactive oxygen species (ROS) and reactive nitrogen species (RNS), being free radicals, are neutralized by endogenous antioxidants in cells and tissues to maintain redox homeostasis [5,6]. This redox homeostasis is disrupted under certain pathophysiological conditions, such as injury, inflammation, genetic mutations, and ischemia/reperfusion, causing oxidative stress, which is associated with a variety of progressive NDs [5,7]. The central nervous system (CNS) can form large amounts of free radicals due to its high oxygen demand and metabolism of neurotransmitters [8,9]. Notably, nerve cells are particularly vulnerable to damage because of the abundance of free radicals and relatively weak antioxidant defenses compared with other organs [10,11]. In addition, oxidative stress causes mitochondrial dysfunction, which is unable to meet the high energy requirements for normal biochemical and physiological functions of neuronal cells, thus leading to neuronal cell death [12]. Thus, maintenance of redox homeostasis is essential for neuronal survival and function.

The gut microbiota comprises the microorganisms that exist in different ecological niches of the gut, including bacteria, fungi, viruses, and protozoa [13]. The gut microbiota significantly affects multiple aspects of host physiology, including the immune system, anti-infection, nutritional metabolism, and nervous system [14,15]. Recently, the gut microbiota has been shown to play an essential role in various biological and physiological processes in the brain, such as glial cell activation, myelination, and neurogenesis [16]. In addition, dysbiosis of the gut microbiota is strongly associated with gastrointestinal diseases, anxiety, depression, metabolic disorders, as well as NDs [17,18,19,20]. These things considered, probiotic strains, such as *Bifidobacterium* and *Lactobacillus*, can produce potential antioxidants, vitamins, and bioactive molecules to maintain redox homeostasis, thereby preventing oxidative stress-related diseases [21,22]. Importantly, the gut microbiota is involved in the communication between the gut and the brain through neurotransmitters and various metabolites [23,24]. The current evidence strongly suggests that the gut microbiota can influence the brain aging process and the initiation and progression of NDs, making the gut–brain crosstalk a promising and exciting research area in neuroscience [16]. It is, therefore, of interest to find novel therapeutic targets and strategies from the perspective of the gut microbiota and redox homeostasis. In this review, we will describe the roles of oxidative stress and the gut microbiota in NDs, as well as the underlying mechanisms by which the gut microbiota affects redox homeostasis. In addition, the potential applications of maintaining redox homeostasis by shaping the gut microbiota to treat NDs will also be discussed.

## 2. Oxidative Stress and NDs

Oxidative stress, the result of an imbalance in the relative abundance of reactive ROS and antioxidants, can create a detrimental state that leads to cellular damage and dysfunction. Increased oxidative stress is able to damage cell membranes, alter protein structure and function, and cause DNA damage [25,26]. Therefore, maintenance of redox homeostasis is essential for cell biological function. As mentioned earlier, the CNS is particularly sensitive to oxidative stress due to its high metabolic rate, relative antioxidant scarcity, and unique structural features [27,28]. In addition, due to the presence of high levels of metal ions and polyunsaturated fatty acids, neuronal cells are more prone to oxidative stress, leading to cell damage and a series of NDs-related events through mitochondrial dysfunction, inflammation, and neuronal death [28,29,30]. In fact, an oxidative stress-induced imbalance in redox homeostasis is still a central component of the pathogenesis of several NDs, such as AD, PD, and MS. The common features among these NDs are ineffective antioxidant defense systems, imbalances of redox homeostasis, mitochondrial dysfunction, neuroinflammation, neuronal loss and degeneration (Figure 1).

Oxidative stress and disruption of cerebral redox homeostasis frequently occur in human NDs. For instance, in the pathology of AD, amyloid β (Aβ) and tau protein aggregates can interact with metal ions and maintain normal cellular signaling [31,32]. Furthermore, previous studies have shown that the high levels of zinc in the neocortical and hippocampal regions of AD patients suggest the vital role of zinc in the maintenance of redox homeostasis in the affected brain regions [33,34]. Notably, accumulated Aβ-induced oxidative stress can inhibit the activity of complex IV, leading to ATP depletion and mitochondrial dysfunction [35,36]. It has been demonstrated that the abnormal aggregation of α-synuclein (α-syn), mitochondrial dysfunction, and excessive oxidative stress are closely related to dopaminergic neuron death during PD progression [37,38,39]. As the main pathogenic factor of HD, soluble and aggregated mutant Htt (mHtt) protein with cytotoxicity induces apoptosis through oxidative stress, resulting in continuous degeneration of neurons [40,41,42]. Interestingly, in patients with NDs, oxidative stress biomarkers such as malondialdehyde and 8-hydroxyguanosine are elevated, and the gene superoxide dismutase 1 (SOD1), which plays an important role in oxidative stress defense mechanisms, is also frequently mutated [43,44].

Despite the advanced understanding of the mechanisms described above, a wide gap remains between this knowledge and the availability of effective therapies. Given that the imbalance of redox homeostasis is one of the key factors in the pathogenesis of NDs, numerous studies have been conducted on the treatment of NDs using various types of antioxidants (Table 1). Overall, most of the clinical trial results of NDs have shown favorable therapeutic effects, especially the alterations of pathological markers and improvements in neurological function, suggesting that antioxidants have great therapeutic potential for NDs. However, more efforts are required to explore novel therapeutic strategies and approaches to achieve broad therapeutic applicability and functional recovery of the nervous system.

## 3. Gut Microbiota, Oxidative Stress, and Neurodegeneration

The human gastrointestinal tract is the largest immune organ and harbors complex and dynamic microbiota [56,57]. Gut microbiota stability can be impacted by several variables, including genetics, lifestyle, nutrition, medications, illness, and age, which in turn have a significant impact on the regulation of metabolism, homeostasis, immunological response, and other processes [58,59]. Therefore, an imbalance in the representation of the gut microbiota may contribute to various diseases, from inflammatory bowel disease to obesity and diabetes, as well as several common NDs, such as AD, PD, and MS (Table 2). In addition, growing numbers of studies have demonstrated that the gut microbiota alters the oxidative/antioxidant balance in the CNS and causes neurodegeneration [60,61,62,63].

### 3.1. Gut–Brain Axis under Physiological Conditions

The brain and the gut microbiota are strictly intertwined and communicate in a variety of ways, including the production of bacterial metabolites, neurotransmitters, and cytokines [74] (Figure 2). Notably, the term “microbiota–gut–brain axis” refers to an interaction between the brain and the gut microbiota that involves four major routes of communication [75,76]. The first route of communication involves the vagus nerve, which connects the enteric nerve system and the brain stem. Recent research indicates that the gut microbiota influences host behaviors such as anxiety, feeding, and depression by activating vagal neurons and altering neurotransmitters such as γ-aminobutyric acid (GABA) and oxytocin in the brain [77,78]. The second important mode that directly or indirectly affects brain activity involves serotonin, which is mainly produced by gut enterochromaffin cells and modulates a variety of physiological processes. Interestingly, increased levels of serotonin and serotonin precursors alleviated depression in a mouse model of depression after treatment with the probiotic Bifidobacterium [79]. Thirdly, the gut microbiota plays an essential role in microglial activation and neuroinflammation. For instance, Luck and colleagues demonstrated that germ-free mice carry more immature microglia than conventional mice, and *Bifidobacterium* spp. can activate microglia through transcriptional activation [80]. In addition, alterations in microglial function were also observed in NDs and other behaviors, suggesting that the gut microbiota mediates effects on NDs through microglia [81]. Notably, the gut microbiota plays a vital role in energy harvest and neuroinflammation, and alterations in the gut–brain vagal pathway may promote obesity. It has been demonstrated that a diet-induced shift in the gut microbiome may disrupt vagal gut–brain communication resulting in microglia activation, increased gut inflammation, and body fat accumulation [82,83]. Finally, the gut microbiota communicates by transferring chemical signals directly to the brain. A previous study indicated that short-chain fatty acids (SCFAs) derived from the fermentation of the gut microbiota had been shown to modulate neuroplasticity in the CNS and improve depressive behavior in mice [62].

### 3.2. Gut Microbiota-Mediated Oxidative Stress and Neurodegeneration

There are four main symbiotic bacteria that are parasitic in the human gut, namely Actinobacteria, Bacteroidetes, Proteobacteria, and Firmicutes. Among them, Firmicutes accounts for the largest proportion, including Streptococcus, Lactobacillus, and Mycoplasma [81]. Recent studies have found that, in the presence of the microbiota, the intestinal epithelium cells produce physiological levels of oxidative stress that affect the composition and function of the gut microbiota. Such alterations in gut microbiota increase the alterations of biomacromolecules reaching the systemic circulation and CNS by directly affecting the permeability of the intestine [84]. Indeed, the gut microbiota can alter cellular oxidative stress status by regulating mitochondrial activity [85]. In addition, gut *Lactobacilli*, *Bifidobacterium*, and *Streptococcus* can produce nitric oxide (NO) in various ways in the gut [86,87]. It is now generally accepted that NO in nanomolar concentrations is neuroprotective, whereas higher concentrations of NO may result in oxidative stress, which is closely related to axonal degeneration, neuroinflammation, and NDs [81]. In addition, certain pathogenic bacteria such as *Salmonella* and *Escherichia coli* are able to produce hydrogen sulfide (H_2_S) in the gut by degrading sulfur-containing amino acids. Furthermore, increased levels of H_2_S alter various host metabolic activities, such as increased lactate, decreased oxygen consumption, decreased ATP production, and elevated levels of proinflammatory compounds, which have been linked to neuroinflammation [88,89,90]. The role of gut microbiota in neurodegeneration is shown in Figure 3.

In-depth studies on the pathogenesis of NDs, including AD, PD, and MS, have mainly focused on the misfolding and aggregation of proteins in neurons. In addition, oxidative stress has also been considered to be closely related to the occurrence and development of NDs, but the exact underlying mechanisms remain unclear. Numerous studies have demonstrated the close association of microbiota-mediated oxidative stress with neurodegeneration. Here, we summarize recent links between the gut microbiota, oxidative stress, and NDs, with a focus on AD, PD, and MS.

#### 3.2.1. Alzheimer’s Disease

AD is the most common ND worldwide with an insidious onset and progressive development [4,91]. It is characterized by progressive impairment of cognition and episodic memory, culminating in the development of dementia [92]. Specific histopathological hallmarks in the brain associated with AD include Aβ plaques, neurofibrillary tangles (NFTs), hyperphosphorylation of tau proteins (tau tangles), and neuronal loss [93,94]. Oxidative stress has been suggested to play an essential role in AD etiology prior to plaque formation, leading to mitochondrial dysfunction in neurons and synapses, as well as Aβ protein production [95,96]. Previous studies have shown that oxidative stress plays a pivotal role in the development of AD. For instance, it has been demonstrated that aggregated Aβ protein stimulates microglia to produce ROS through positive feedback on Aβ plaque deposition [91]. In addition, tau protein aggregation in neurons leads to reduced NADH-ubiquitin reductase activity, leading to oxidative stress and mitochondrial dysfunction [97]. Interestingly, ROS can affect the activity of stress kinases, such as the phosphorylation-c-Jun N-terminal kinase 1 (p-JNK) pathway, which is associated with neuronal cell death due to tau hyperphosphorylation and accumulation of Aβ [98]. There is ample evidence that the oxidation of nucleic acid species in the AD brain is dominated by the mitochondrial genome, and lipid peroxidation results in the production of certain cytotoxic agents, such as 4-hydroxyalkenals [99,100,101].

Recent studies have shown that the gut microbiota plays a significant role in the pathogenesis of AD [102]. Dysregulation of the gut microbiota leads to oxidative stress, inflammation, disruption of the blood–brain barrier, activation of the immune system, neurofibrillary tangles, and Aβ plaques followed by neurodegeneration [21,103]. There are numerous bacteria in the human gut that play a vital role in the etiology of AD, including *Staphylococcus aureus*, *Escherichia coli*, *Salmonella*, *Mycobacterium*, *Klebsiella pneumoniae*, and *Streptococcus*, which promote the production and aggregation of the Aβ protein in the enteric nervous system [104,105]. Interestingly, in the APP_SWE_/PS1_ΔE9_ transgenic mouse model of AD chronically treated with broad-spectrum combination antibiotics, the gut microbiome of transgenic mice shifted toward proinflammatory bacteria, with a decrease in amyloid plaque deposition and neuroinflammation [66,106]. Additionally, microbial amyloid protein, produced by coccus-shaped bacteria, is able to activate the innate immune system and triggers responses by Toll-like receptors (TLRs) and cluster of differentiation 14 (CD14), resulting in inadequate recognition of misfolded Aβ and decreased Aβ clearance, followed by the production of cytokines leading to intestinal disturbances [107]. Notably, age-related reductions in gut microbial diversity are also implicated in AD. It has been demonstrated that with growing age, there is an increase in Proteobacteria and a decrease in Bifidobacterium spp., which results in interference in lipid metabolism and a failure to maintain hippocampal plasticity as well as memory functions [108,109].

Another possible connecting link between the gut microbiota and microbiota-mediated cerebral amyloid accumulation involves a cross-seeding mechanism of microbial amyloid (i.e., promoting misfolded aggregation of amyloid from one protein to another) in a manner similar to the reproduction of prions [107,110,111]. Notably, distinct amyloid conformations interact with cellular targets to produce various toxicities, which may explain the different AD phenotypes [112]. Given the multiple roles of gut microbiota dysbiosis in the pathogenesis of AD, modulation of AD through dietary and gut microbiota interventions may be potential therapeutic strategies, which will be discussed in detail later.

#### 3.2.2. Parkinson’s Disease

PD, the second most common ND after AD, is a long-term neurological disorder that causes both motor and non-motor symptoms [113,114,115]. Motor symptoms include resting tremors, akinesia, muscular rigidity, postural instability, and gait abnormalities [116,117,118]. Non-motor symptoms include anxiety, depression, autonomic dysfunction, cognitive decline, and sleep disturbances [119,120]. The hallmarks of PD are loss of dopaminergic (DA) neurons and abnormal accumulation of α-syn within the cytoplasm of nerve cells called Levy bodies [121,122,123,124]. Notably, oxidative stress, mitochondrial dysfunction, dopamine metabolism, abnormal protein aggregation, and the gut microbiota are associated with the pathogenesis of PD [125,126,127]. As one of the main pathogenic factors of PD, oxidative stress has been linked to α-syn protein aggregation and degeneration in DA neurons [98,121,128]. For instance, analysis of the postmortem brain tissue in PD showed that oxidative stress degenerates DA neurons, reduces the levels of glutathione (GSH), increases the levels of oxidative stress markers, and stimulates lipid, DNA, and RNA oxidation [129,130]. Additionally, Tong and colleagues have demonstrated that oxidative stress in DA neurons can activate the p38 mitogen-activated protein kinase pathway, ultimately leading to neuronal apoptosis [131]. Interestingly, in the 6-hydroxydopamine-induced PD model in mice, Antrodia camphorata polysaccharide reduced ROS by increasing the expression and activity of antioxidant enzymes, ultimately attenuating the damage of DA neurons in the substantia nigra and improving motor performance [132].

Notably, PD patients often present with gastrointestinal dysfunction, which suggests that the imbalance of the gut microbiota is one of the causes of triggering or aggravating PD [133,134]. Indeed, gut inflammation, early accumulation of α-syn, increased intestinal permeability, and constipation problems are common in PD patients, again demonstrating the critical role of the gut microbiota in PD [81,135]. It has been demonstrated that the disruption of gut microbiota leads to oxidative stress through overstimulation of the immune system, which in turn activates intestinal neurons and intestinal glia cells, leading to increased misfolding and accumulation of α-syn in the CNS [136,137]. In addition, coming to the role of the gut microbiota, toxins and microbial products produced by certain pathogenic bacteria are able to cause mitochondrial dysfunction in intestinal cells and the CNS, which is directly associated with PD pathogenesis [138]. In line with this, it is proposed that the pathogenic bacterium *E. coli* can produce an amyloid protein called curli, which promotes the accumulation of α-syn protein in the brain and causes motor defects in mice [139]. Conversely, when treated with gut-restricted amyloid inhibitors, these mice showed significant improvements in constipation and motor function, suggesting the role of the gut microbiota in the etiology of PD symptoms [140]. Interestingly, when the gut microbiota from PD patients was transplanted into a germ-free α-syn overexpressed mouse model, a similar pattern of physical injury to PD patients was observed, suggesting the vital role of the gut microbiota in PD. As discussed above, decreased production of hydrogen (H_2_) by the gut microbiota has been proposed as one of the essential factors in PD [141]. According to a recent study, 50% H_2_ saturated water was successful in preventing nigrostriatal degeneration in PD rats and reducing the oxidative stress markers in the 1-methyl-4-phenyl-1,2,3,4-tetrahydropyridine (MPTP) mouse model [142]. Taken together, these observations suggest a critical role for gut microbiota in PD, and intervention through gut microbiota is expected to provide promising strategies for the prevention and treatment of PD.

#### 3.2.3. Multiple Sclerosis

MS is an immune-mediated chronic inflammatory and central nervous system demyelinating disease with a complex and unclear pathogenesis [143]. It has been established that the pathogenesis of MS involves both genetic and environmental factors. The most extensively accepted hypothesis is that autoreactive B and T cells cause axonal and myelin damage, as well as neurodegeneration [144,145]. The major neuropathological hallmarks of MS pathology are inflammation and degeneration of both white matter and gray matter [146]. However, the development of MS may be influenced by a combination of internal and external factors, ultimately leading to immune dysregulation.

Growing evidence suggests that the imbalance of redox homeostasis plays a vital role in the pathogenesis of MS. For instance, it has been demonstrated that the excessive generation of ROS, mitochondrial dysfunction, and impairment of antioxidant defense systems play important roles in the pathogenesis of MS [147]. Notably, ROS has been shown to be a mediator of axonal injury and demyelination in both MS patients and animal models of MS. In addition, oxidative stress mediates mitochondrial dysfunction in MS patients and leads to CNS energy failure in MS-susceptible individuals [148,149]. Furthermore, recent studies have demonstrated that the gut microbiota has a significant impact on MS and can be influenced by external factors [150]. For instance, Cosorich et al. have demonstrated that T helper 17 (TH17) cells, key players in MS, originate in the gut and that increased TH17 cell frequency is associated with specific alterations of the gut microbiota in MS patients [151]. Interestingly, transplantation of the MS microbiota in a mouse model resulted in an increased incidence of autoimmune encephalomyelitis, leading to an exacerbation of MS symptoms [152,153]. Additionally, diets have been shown to affect the balance of the gut microbiota and indirectly influence the development of MS [154]. Moreover, dietary studies in MS patients suggest that dietary interventions supplemented with vitamin D in a low-calorie diet have a positive effect on alleviating chronic inflammatory symptoms in MS [155]. Recently, intermittent fasting was introduced into the treatment of MS due to its availability of abundant gut microbiota as well as the secretion of glutathione and leptin [156]. All these studies have shown that modification of the gut microbiota can be considered a promising therapeutic strategy for MS.

## 4. Gut Microbiota in Neuroprotection

The complex gut microbiota and microbiota–host interactions may directly and indirectly affect the oxidative state of the CNS by producing numerous metabolites such as absorbable vitamins, SCFAs, polyphenols, diffusible antioxidants, and oxidant gases [60]. Notably, the gut microbiota is also able to optimize dietary energy harvest, influence the permeability of the blood–brain barrier and the intestinal barrier, modulate the immune response, and prevent the extensive colonization of pathogens [157,158]. As a component of the parasympathetic nervous system, the vagus nerve can sense intestinal metabolites, communicate with the CNS, and integrate into the central autonomic network to generate specific responses. For instance, under stressful conditions, the vagus nerve can be suppressed, with deleterious effects on the gastrointestinal tract and microbes, such as inflammatory bowel disease (IBD) and irritable bowel syndrome (IBS) due to dysbiosis [157]. Furthermore, the beneficial gut microbiota produces a large number of CNS neurotransmitters such as serotonin, dopamine, and γ-aminobutyric acid, which modulate enteric nervous system (ENS) activity and may be associated with their respective levels within the CNS depending on gut permeability and the blood–brain barrier [10].

With our increasing understanding of the essential role of gut microbiota and oxidative stress in NDs, there is a growing need to develop therapies based on antioxidant strategies to treat NDs. Although antioxidants have a potent therapeutic effect on certain diseases, the therapeutic effects of antioxidants for NDs are still limited and require deep mechanistic understanding [159]. As discussed above, the dichotomous role of gut microbiota has been observed. On the one hand, the gut microbiota is closely related to the underlying pathogenesis of neurodegeneration. On the other hand, the gut microbiota and its metabolites modulate numerous NDs-related pathways, suggesting their potential therapeutic role in neuroprotection (Figure 4). Therefore, maintaining neuronal health by modulating gut microbiota homeostasis holds great promise. Here, we describe the role of metabolites from the gut microbiota as well as the antioxidative and anti-inflammatory probiotics in neuroprotection.

### 4.1. Interactions of Gut Microbiota with Host and Dietary Molecules

Many host molecules, such as bile acids and steroid hormones, are produced via multistep biosynthetic pathways and can interact with the gut microbiota to have beneficial or deleterious effects on the host system. Bile acids are produced in the liver and released in the gut, mainly associated with the dissolution of lipids and fat-soluble vitamins. They also play a vital role in the physiological and pathological processes of the CNS [160]. Bile acids such as taurine deoxycholic acid (TUDCA) and ursodeoxycholic acid (UDCA) have been shown to be neuroprotective without cytotoxicity [161,162]. Moreover, Cuevas et al. revealed that pretreatment with TUDCA protected against dopaminergic neuronal damage, attenuated protein oxidation and autophagy, and also prevented α-syn aggregation [163]. Likewise, UDCA was found to convey neuroprotection in drosophila and mammalian models of charged multivesicular body protein 2B (CHMP2B) Intron 5 (CHMP2BIntron5) induced frontotemporal dementia (FTD) [164]. Indeed, the gut microbiota plays a vital role in the conversion of primary bile acids to secondary bile acids and can alter their solubility, nuclear receptor binding, and blood circulation [165]. Notably, altering bile acid levels and properties by modulating the gut microbiota may have neurodegenerative and neuroprotective effects. For example, alterations in secondary bile acid levels have been found in human and mouse models of a variety of NDs such as AD, PD, and MS [166,167,168]. Furthermore, a previous study indicated that TUDCA might exert neuroprotective effects by inhibiting inflammatory responses and oxidative stress in microglia [169]. However, extensive research is still required to elucidate the specific mechanisms and potential roles of the gut microbiota in manipulating bile acids. Other common host molecules are steroid hormones, which are crucial for brain physiology and function [170]. Through degradation and activation pathways, the gut microbiota can modulate the level of steroid hormones [171]. Moreover, it has been demonstrated that both androgen and estrogen secretions are affected by the gut microbiota. Actually, most gut bacteria are able to metabolize estrogen and can also promote estrogen to undergo oxidation–reduction reactions [172,173]. Interestingly, estrogen influenced by the gut microbiota is neuroprotective and shows anti-inflammatory and differentiated effects in nerve cells [174,175]. Accordingly, an altered gut microbiota leads to low levels of estrogen, which leads to neuroinflammation and neurodegeneration [176,177].

Additionally, dietary molecules such as amino acids, dietary fibers, and polyphenols are also inextricably linked to the gut microbiota. Dietary amino acids are normally metabolized by the gut microbiota, and the resulting dietary amino acids affect the CNS [178,179]. Norepinephrine can be produced by the gut microbiota in the millimolar range, and it protects neurons from H_2_O_2_-induced death by increasing the supply of GSH from astrocytes [180]. In addition, indole propionic acid, an indole derivative, is a product of tryptophan metabolism by the gut microbiota and acts as an antioxidant that reduces neuroinflammation and attenuates AD pathology [181,182]. Notably, as a branch of tryptophan metabolism, the disturbance of the kynurenine (KYN) pathway was found to affect memory, anxiety, and stress-related behavior and promote inflammatory responses and neurotoxicity, suggesting the neuroprotective and anti-inflammatory role of KYN in NDs [183,184]. Furthermore, as one of the metabolites of arginine, agmatine is involved in the major processes of synaptic plasticity and memory formation and has therapeutic effects on a variety of NDs [185,186]. Meanwhile, agmatine has been shown to stimulate the NRF2 signaling pathway to reduce the production of ROS and protect neuronal cells from oxidative stress-induced damage [187,188]. In addition, undigested dietary fiber in the body is converted into SCFAs through anaerobic fermentation of gut microbiota, which can not only provide energy but also affect the development and function of the CNS directly or indirectly [189]. It has been reported that SCFAs maintain redox homeostasis in the brain via regulating microglia homeostasis, thus attenuating neuroinflammation in AD and PD [190]. Interestingly, transplantation of fecal microbiota from wild-type mice into the mouse model of PD, combined with butyrate treatment, significantly improved PD symptoms [191]. Butyrate has also been shown to affect the neuroinflammation and the cellular oxidative status of astrocytes [192]. Indeed, fecal transplantation is seeing increased use for the clinical management of neurodegenerative diseases and a number of clinical trials have been undertaken [193,194]. In summary, SCFAs obtained from dietary fiber have great therapeutic potential for NDs. Another group of bioactive molecules in plants is the polyphenols, which can be classified into phenolic acids, flavonoids, and tannins. Due to their special structure, they are able to scavenge free radicals and have been widely applied as antioxidants in the treatment of NDs [195]. Many dietary polyphenols have been shown to be actively converted by the gut microbiota to phenolic acids such as 3-hydroxybenzoic acid and 3-(3-hydroxyphenyl) propionic acid, which inhibit A aggregation and the progression of AD [196]. Similarly, proanthocyanidins (PA) can attenuate oxidative stress in dopaminergic neurons by inhibiting p38, ERK, and JNK signaling pathways, which may provide a new perspective for PD therapy [197]. Taken together, the above studies suggest the therapeutic potential of the interactions of gut microbiota with host and dietary molecules for NDs.

### 4.2. Vitamins from Gut Microbiota in Neuroprotection

Because the human body lacks the biosynthetic capacity for most vitamins, they must be exogenously supplied to meet demand. Although vitamins are present in various foods, inadequate food intake and poor dietary habits can still cause vitamin deficiencies [198]. Notably, the gut microbiota is a rich source of vitamins, especially vitamins B and K, which are required by both the host and certain gut microbiota [199,200]. In the gut, lactic acid bacteria, *Bacillus subtilis*, and *E. coli* produce vitamin B2 (riboflavin). B6 (pyridoxine, pyridoxamine, and pyridoxal) is produced by pyridoxine, pyridoxamine, and pyridoxal, and vitamin K is produced by *Escherichia coli*, *Propionibacterium*, and *Eubacterium* [198,200]. Although dietary vitamins are absorbed primarily from the small intestine, vitamins derived from gut microbiota are taken up in the distal colon and perform various important functions in the body, particularly in neuroprotection. For instance, it has been demonstrated that vitamin K deficiency is strongly associated with the pathogenesis of AD and that increasing the intake of dietary vitamin K is helpful in improving memory function in elderly patients [81]. Additionally, vitamin K2 exerts potent antioxidant properties by inhibiting the activation of the P38 signaling pathway, ROS generation, and the activity of caspase-1, thereby restoring mitochondrial membrane potential, demonstrating its potential for PD treatment [201]. Interestingly, supplementation of B vitamins such as B6, B9, and B12 can slow the shrinkage of specific brain regions associated with cognitive decline in AD [202]. The above studies suggest that vitamins B and K play an important role in improving neuronal health. However, further constructive research is required to demonstrate the neuroprotective potential of vitamins produced by the gut microbiota.

### 4.3. The Effect of Probiotics in Neuroprotection

Probiotics are non-pathogenic living microorganisms known for their beneficial effects on health; they include *Lactobacillus*, *Streptococcus*, *Propionibacterium,* and *Bifidobacterium* [203,204,205]. Notably, when the gut microbiota is perturbated leading to certain diseases, probiotic treatment can restore the gut microbiota and ensure the normal functions of the body, which indicates the vital role of probiotics in health [206]. As the most common probiotics, *Lactobacillus* and *Bifidobacterium* can alter the composition and quantity of gut microbiota, improve intestinal barrier function, regulate mood and cranial nerve status, and confer resilience to stress [77,207,208,209,210]. Growing evidence indicates the positive effect of probiotics on redox homeostasis in NDs. Probiotic strains dominated by *Bifidobacterium* and *Lactobacillus* are able to combat excess free radicals in the form of ROS in the body by producing antioxidants, vitamins, and other bioactive molecules, thereby preventing oxidative stress-related diseases, especially NDs [22,211].

Research on probiotics in NDs in recent years has focused on their antioxidant effects. Probiotics affect brain function and the progression of NDs via their ability to modulate ROS-producing enzymes, chelate metal ions, activate antioxidant pathways, and produce antioxidant metabolites [212,213]. For instance, it has been shown that probiotic consumption has a positive effect on cognitive function and certain metabolic statuses in AD patients [214]. In addition, Akbari et al. have revealed that *L. plantarum* can reduce malondialdehyde and stimulate the activity of SOD and GPX, thereby scavenging hydroxyl radicals in mice with d-galactose-induced oxidative stress [214]. Interestingly, in addition to inhibiting ROS production by NETs to exert antioxidant effects, *L. rhamnosus* was found to have antidepressant and antianxiety properties, possibly related to neuroactive substances of bacterial origin [215]. Notably, the gut microbiota can exert antioxidant activity through their anti-inflammatory effects, which is considered to improve the symptoms of a wide range of disorders [216,217]. Wu et al. have revealed that an important strain of *Lactobacillus*, *L. fermentum*, could prevent ROS formation by stimulating the production of IL-10 [218]. In addition, the combination of *L. mucosae AN1* and *L. fermentum SNR1*, with strong antioxidant activity, can decrease the level of proinflammatory cytokines, increase the level of anti-inflammatory cytokines, and inhibit related mediators in the gut, thereby alleviating oxidative stress, inhibiting the activation of inflammation-related pathways and maintaining redox homeostasis [217,219]. Similarly, a previous study indicated that a significant reduction in the abundance of anti-inflammatory bacteria with an increase in the abundance of proinflammatory bacteria is possibly associated with cognitive impairment in AD patients [68]. Taken together, the above studies suggest that probiotics may serve as a potential therapeutic intervention for NDs.

## 5. Shaping the Gut Microbiota to Maintain Redox Homeostasis for NDs Treatment

Given the close interactions between oxidative stress and the gut microbiota, targeting the gut microbiota and redox homeostasis may represent potential therapeutic strategies for treating NDs. Actually, numerous studies and clinical trials have comprehensively revealed the positive role of gut microbiota in the treatment of NDs. For instance, sirtuin-1 (SIRT1), a probiotic bacterial protein, has been shown to have neuroprotective effects, and alterations in the expression and activity of SIRT1 protein are closely associated with Aβ and tau accumulation in the cerebral cortex in both animal models of AD and human AD patients. When treated with probiotic supplements, the SIRT1 pathway is activated to exert antioxidant effects [81,220,221]. In addition, prolonged diet supplementation with a Lactobacillus strain upregulates the expression of brain-derived neurotrophic factor (BDNF) in the hippocampus, thereby preventing age-associated cognitive decline [222]. Similarly, Lactobacillus paracasei PS23 supplements can prevent aging-related neurological damage and ameliorate cognitive dysfunction, possibly by increasing the activity of antioxidant enzymes in the hippocampus and modulating microbiota–gut–brain axis communication [223]. Indeed, Bifidobacterium bifidum ATCC 29,521 exerts a beneficial effect on murine gut microbiota and redox homeostasis and could be a potential bioresource antioxidant in effective functional foods [224].

Notably, growing evidence indicates the efficacy of this novel therapeutic strategy for treating or alleviating NDs. A previous study indicated that probiotic mixtures consisting of *S. thermophilus*, *L. plantarum*, *B. breve*, and other probiotics in the treatment of early AD in transgenic mice could improve cognitive dysfunction and reduce brain damage by inhibiting Aβ plaque formation and altering gut microbiota [211]. Moreover, *Bifidobacteria* and *Lactobacilli* strains have been shown to produce vitamins and bioactive molecules as antioxidants, which play a major role in the pathology of PD [225]. Furthermore, it has been demonstrated that various species of *Lactobacillus* and *Bifidibacterium* are able to prevent the progression of the experimental autoimmune encephalomyelitis (EAE) animal model of MS and improve clinical symptoms [226,227,228]. Indeed, clinical trials indicate that Streptococcus spp., Lactobacillus spp., and Bifidobacterium spp. supplementation is able to reverse MS-induced variations in the gut microbiota composition in MS subjects [229]. The current methods of intestinal flora operation, including cecal fistula, have a broad application prospect. Notably, a previous study shows that 3D-printed cecal fistula implantation, which, through the body wall and into the cecum of rats to obtain long-term access to the gut microbiome, is an effective procedure that allows long-term and minimally invasive access to the gut microbiome [230]. Taken together, modification of the gut microbiota while maintaining redox homeostasis could be a promising therapeutic strategy for NDs in the future.

## 6. Conclusions and Perspectives

Overall, growing lines of evidence support the close link between redox homeostasis and the gut microbiota in the development of NDs. In this regard, probiotic supplements have a positive effect on redox imbalance and damaged gut microbiota, effectively exerting neuroprotective effects. Therefore, shaping the gut microbiota to maintain redox homeostasis appears to be a novel and effective therapeutic strategy for the treatment of NDs.

However, the gut microbiota is a vast and diverse reservoir of microorganisms, and more investigation is required to examine and characterize the role of the gut microbiota, as well as the interactions between oxidative stress and the microbiota–gut–brain axis. As high-throughput sequencing technologies continue to evolve, as well as multi-omics approaches to machine learning, they can help unravel the intricate networks of interactions involved. For instance, mass spectrometry-based metabolomics, single-cell RNA sequencing, and spatial transcriptomics should enable the identification of associated metabolic pathways and untangle the connections involved in the microbiota–gut–brain axis, contributing to the development of precision medicine. Bioinformatics tools will be essential, to mine the resulting data, enabling drugs to be designed that target the specific gut microbiota that mediate oxidative stress and the production of harmful metabolites, and exploring the neuroprotective and neurodegenerative roles of the gut microbial metabolites. Accordingly, further studies will be conducted to reveal how gut microbiota-mediated oxidative stress contributes to the prevalence of NDs and reveal novel therapeutic strategies to combat such conditions.

## Figures and Tables

**Figure 1 antioxidants-11-02287-f001:**
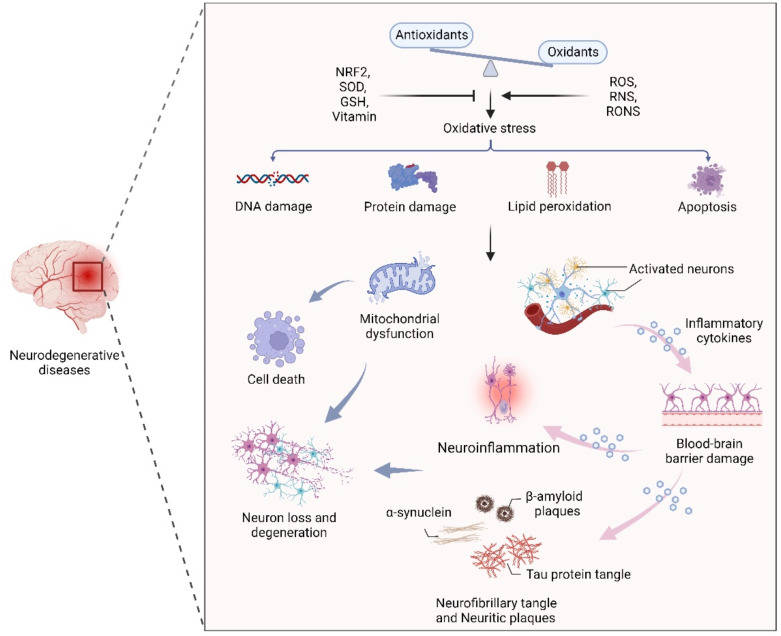
The effect of oxidative stress in neurodegenerative diseases. An oxidant/antioxidant imbalance leads to oxidative stress, which causes DNA and protein damage, lipid peroxidation, and apoptosis. Dysfunctional mitochondria and activated neurons secrete inflammatory cytokines that cross the blood–brain barrier, leading to inflammation, α-synuclein, β-aggregation, and neuronal plaque accumulation in neurons, leading to neuron loss and degeneration.

**Figure 2 antioxidants-11-02287-f002:**
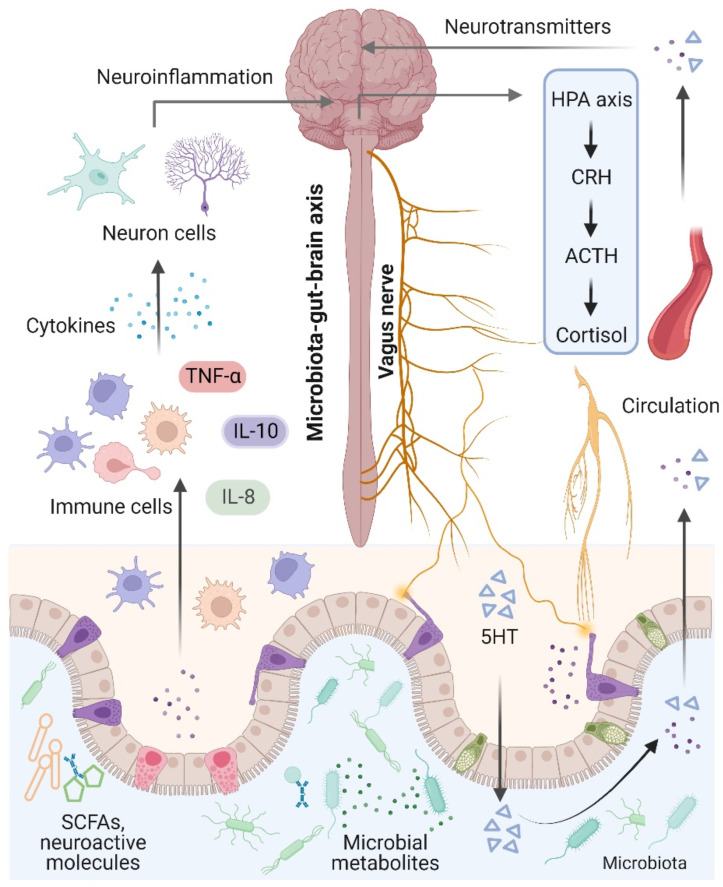
Microbiota–gut–brain axis. The brain and gut communicate through neural, metabolic, endocrine, and immunological pathways. The brain influences gut health through the vagus nerve, the hypothalamic–pituitary–adrenal (HPA) axis, and systemic circulation. Signals from the gut, including short-chain fatty acids (SCFAs), neurotransmitters, and amino acids, modulate brain function via neuronal cells, the immune system, and endocrine mechanisms.

**Figure 3 antioxidants-11-02287-f003:**
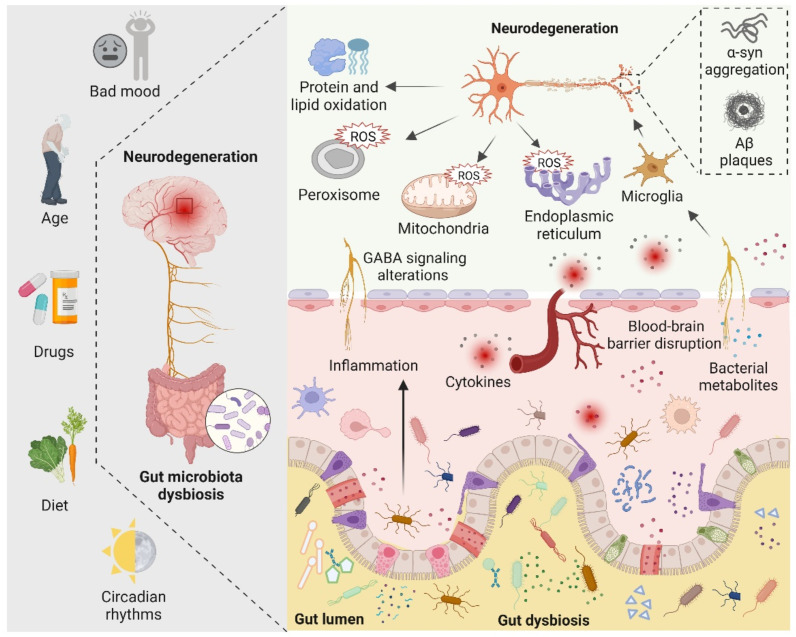
The role of the gut microbiota in neurodegeneration is depicted schematically. Bad mood, increasing age, drugs, dietary changes, and circadian rhythms can disrupt gut microbiota homeostasis. When gut dysbiosis occurs, beneficial bacteria in the gut are transformed into pathogenic bacteria, producing a large number of harmful metabolites and proinflammatory molecules, resulting in increased blood–brain barrier permeability and peripheral inflammatory responses, thereby aggravating oxidative stress in the brain. At the same time, dysbiosis can induce bad mood. Increased levels of ROS in neuronal mitochondria, endoplasmic reticulum, and peroxisomes, increased protein and lipid oxidation, and accumulation of neurotoxic proteins lead to neurodegeneration.

**Figure 4 antioxidants-11-02287-f004:**
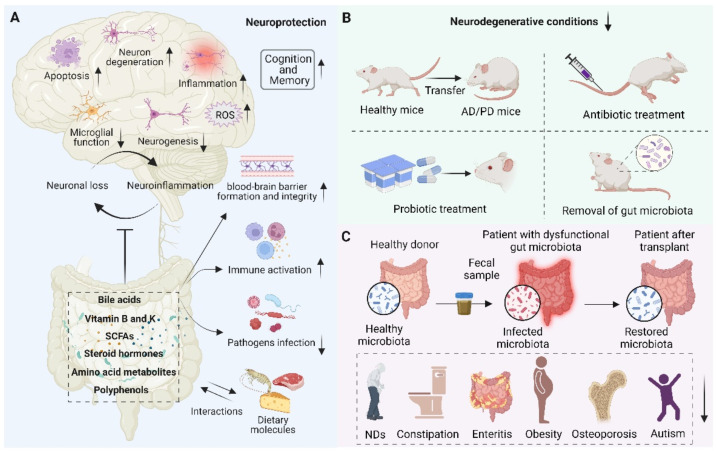
Schematic representation of the role of the gut microbiota in neuroprotection. (**A**) Beneficial metabolites such as bile acids, vitamins, short-chain fatty acids (SCFAs), steroid hormones, amino acid metabolites, polyphenols, etc., released by the gut microbiota and interacting with dietary molecules, can promote blood–brain barrier formation and integrity, reduce inflammation, reduce oxidative stress, reduce neuronal apoptosis, activate immune response, prevent pathogen infection, and thus play an important role in neuroprotection. (**B**) Fecal microbial transfer (FMT), antibiotic and probiotic treatment, and removal of gut microbiota were found to decrease neurodegenerative conditions and reduce the pathophysiology of NDs. (**C**) The scheme of transferring the gut microbiota from a healthy donor to a patient with dysfunctional gut microbiota and restoring the microbiota, thereby improving human health, such as treating NDs, constipation, enteritis, obesity, osteoporosis, and autism.

**Table 1 antioxidants-11-02287-t001:** Antioxidants with therapeutic effects on neurodegenerative diseases.

Antioxidants	Therapeutic Target	Mechanism	Reference
Luteolin	PD	Increased dopamine absorption	[45]
Selenium	AD	Degradation of Aβ plaques	[46,47]
Curcumin	PD	NRF2 activation	[48]
α-Tocopherol	AD	Aβ plaque degradation	[49]
Quercetin	AD, PD	Hydroxyl radical scavenging	[50]
Ginsenosides	AD	Inhibition of Aβ aggregation	[51]
PLGA NPs	AD, PD, MS	Protection against oxidative stress	[52]
Macrophage-derived exosomes	PD	Protection against oxidative stress and inflammation	[53]
Coenzyme Q10	AD	Reduction of oxidative stress and senile plaques	[54]
Ferulic acid	AD	Inhibition of neuronal oxidative stress	[55]

**Table 2 antioxidants-11-02287-t002:** Alterations in the gut microbiota composition in various neurodegenerative diseases.

Neurodegenerative Disease	Experimental Subject	Gut Microbiota	Reference
AD	Fecal samples from AD	Firmicutes, Bifidobacterium ↓ Bacteroidetes ↑	[64]
	Symptomatic Tg2576 mice	Firmicutes, Bacteroidetes, Lactobacillus ↑	[65]
	Fecal samples from AD patients	Ruminococcacea ↑ Lachnospirace ↓	[66]
	Male patients with AD	Bacteroidetes, Blautia ↑ Firmicutes, Bifidobacterium ↓	[67]
	Amyloid-positive patients	Escherichia, Shigella ↑ Eubacterium rectale ↓	[68]
PD	Patients with PD	Enterobacteriaceae, Serratia ↑ Blautia, Coprococcus, Lachnospiraceae ↓	[69]
	16S microbiome datasets	Akkermansia, Lactobacillus, Bifidobacterium ↑ Faecalibacterium, Lachnospiraceae ↓	[70]
	Patients with PD	Butyricicoccus, Clostridium ↑ Shigella, Lactobacillus ↓	[71]
MS	Patients with MS	Caproic acid, producers ↑ Butyric acid, producers ↓	[72]
	Patients with MS	Patescibacteria ↑ Lachnospiraceae, Ruminococcaceae ↓	[73]

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
