# Peer review of "New Insights into the Gut Microbiota in Neurodegenerative Diseases from the Perspective of Redox Homeostasis"

_antioxidants, 2022, doi:10.3390/antiox11112287_

Round 1

Reviewer 1 Report

In this paper, Wang and colleagues reviewed the role and the contribution of oxidative stress and gut microbiota in neurodegenerative diseases and the underlying mechanisms by which the gut microbiota affects redox homeostasis in the brain, leading to neurodegenerative diseases. 

I read with great pleasure the manuscript. Overall, the paper is interesting, well-organized, and nicely written. The Figures are very clear and helpful. The title reflects the subject matter. The English language and style are fine; in any case, a minor spell check is required throughout the manuscript. 

One minor suggestion is the possibility to add an abbreviation list at the end of the paper.

Some other minor issues are reported below, that need to be fixed.

Lines 97-98: Please consider changing the sentence as follows: “Oxidative stress and disruption of cerebral redox homeostasis frequently occur in human NDs.”

Lines 102-104: Please consider removing “in AD” at the end of the sentence.

In Figure 1: please considering to better detail β-amyloid plaques, Tau, a-synuclein.

Lines 144.145: please considering to add some references, For example: Powell et al (2017). Nat. Rev.

Gastroenterol. Hepatol. 14, 143–159; Natale et al (2021) Life 11:732. 

Lines 231-232: it would be worthy to specify the transgenic mouse model, for the sake of clarity.

Line 235: “by a bacterial population”. Please specify which one.

Line 351: please correct “NDS”.

Line 424: [189][190]. Please correct.

Please revise H2 and H2O2 throughout the text.

Line 517: amyloid β (Aβ), amyloid plaque, amyloid-β, Aβ plaque. Please use the very same name throughout the text.

Reviewer 2 Report

Re: Manuscript ID: antioxidants-1963658

This is a well written review that deals with the intriguing role of the microbiota-gut-brain axis in the pathogenesis of neurodegenerative diseases. In particular, the role of oxidative stress was examined. The different points are well balanced and developed. Some minor points of criticism have been raised.

Points of criticism

Bacterial strains in italics.

References are not in the style of the journal.

Other comments were reported directly on the manuscript (see pdf attached).

Reviewer 3 Report

In the manuscript titled “New insights into the gut microbiota in neurodegenerative diseases from the perspective of redox homeostasis”, Authors aimed to review recent progress regarding the role of oxidative stress and the gut microbiota in neurodegenerative diseases. This is an important topic because increasing the understanding of mechanisms by which microbiome modulates progression of neurodegeneration could enable the methods of microbiome manipulations to combat neurodegenerative diseases. The following comments need to be addressed in the presented study:

Abstract:

The significance sentence should be added at the end of the abstract.

Introduction:

The last paragraph of the introduction on page 2 should be expanded by introducing current methods of microbiome manipulations presented in the recent review on the role of microbiome in neurodegenerative diseases ( https://www.mdpi.com/2072-6643/13/1/74 ).

Chapter 3.1. Gut-brain axis under physiological conditions:

In this chapter the role of microbiome in gut-brain axis should be expanded by presenting the recent studies regarding influence of gut microbiome on neuroinflammation ( https://www.ncbi.nlm.nih.gov/pmc/articles/PMC5382806/ ; https://www.sciencedirect.com/science/article/pii/S0031938416309489  ).

Chapter 5. Shaping the gut microbiota to…:

The chapter should review the current methods of gut microbiota manipulations including the cecal fistula ( https://www.mdpi.com/2072-6643/13/12/4515 ).

General comments:

It would be beneficial for the review to add the chapter regarding diet-microbiome-neurodegeneration/oxidative stress studies.

Round 2

Reviewer 1 Report

The Authors have addressed all the concerns raised by the Reviewer. I have no more academic questions.

Reviewer 2 Report

The authors amended all the comments and the manuscript is now suitable for publication.

Reviewer 3 Report

All comments addressed satisfactory.